# A Meteor Detection Algorithm for GWAC System

Yicong Chen [1], Guangwei Li [2,*], Cuixiang Liu [1,*], Bo Qiu [1,*], Qianqian Shan [1] and Mengyao Li [1]

1  School of Electronic and Information Engineering, Hebei University of Technology, Tianjin 300401, China; 202121902006@stu.hebut.edu.cn (Y.C.); 202221902012@stu.hebut.edu.cn (Q.S.); 202221902026@stu.hebut.edu.cn (M.L.)
2  Key Laboratory of Space Astronomy and Technology, National Astronomical Observatories, Chinese Academy of Sciences, Beijing 100101, China
*  Correspondence: lgw@bao.ac.cn (G.L.); liucuix@hebut.edu.cn (C.L.); qiubo@hebut.edu.cn (B.Q.)

**Abstract:** Compared with the international meteor surveillance systems, the ground wide angle camera (GWAC) system exhibits characteristics such as images with the resolution of 4K × 4K and single-site observation. These characteristics present challenges for meteor detection in the GWAC system. Consequently, this paper proposes a new meteor detection algorithm for the GWAC system on the base of the solely mini-GWAC system data algorithm. The new algorithm consists of the following key steps: (1) to compare differences between adjacent frames, applying block-based image binarization thresholds, and incorporating median filtering to reduce noise; (2) to adopt the probabilistic Hough transform (PHT) to identify moving objects and cluster them based on the origin moment of the line segments, while assessing the credibility of clustering; (3) to introduce the so-called maximum disappearance frame for moving objects in the tracking algorithm, enhancing the ability to track multi-frame moving objects. The utilization of the line segment inclination angle of the moving object as the direction of movement facilitates the tracking of multiple moving objects, thereby reducing the probability of mistakenly selecting single-frame moving objects; (4) to leverage the light curves of single-frame moving objects to select meteors to enhance the accuracy of meteor detection. Comparative experiments demonstrate that our proposed algorithm processes each frame image in just 0.39 s, achieving an accuracy of 89.8% in the dataset of 5856 adjacent frames. The experimental results indicate that the algorithm achieved an accuracy of 90.27% when applied in the meteor detection of the image data captured by the GWAC system from Dec. 10th to 19th in 2019 and 2021, obtaining excellent detection results.

**Keywords:** meteor detection; GWAC; moving objects tracking; light curve

## 1. Introduction

The origin of meteor astronomy can be traced back to the 19th century [1–3], when scientists began to systematically study and observe meteors. For example, understanding the characteristics of meteors can provide information about the interplanetary dust environment surrounding Earth and how it has evolved during the solar system's evolutionary process. Meteor automation processing is a critical technology that enables efficient detection, tracking, and analysis of meteors. Scientists can obtain accurate meteor trajectories and mass characteristics, thereby promoting research on the origin and physical properties of meteors [4].

Most of the existing international meteor networks detect meteors through temporal and spatial correlations [5], primarily utilizing techniques such as temporal difference, pixel averaging, thresholding, and Hough transform (HT). MeteorScan [6] compares frame differences between adjacent time image pairs and performs HT recognition and matched filtering to reduce false detection probability. MetRec [7] reduces noise by averaging pixels in a 2 × 2 fashion and downsamples the image size by a factor of four. It employs multi-frame region of interest (ROI) tracking to detect meteor, but this results in higher

computational complexity. In contrast, UFOcapture [8] does not utilize mean differencing and spatial-temporal correlation. It applies a $5 \times 5$ spatial filter with frame differencing that is then masked and thresholded. ASGARD [9,10] performs real-time detection by comparing the pixels of the current video frame with the previous 10 frames, counting the number of pixels with increased brightness that exceed a set threshold to reduce noise interference.

There are a number of meteor surveillance systems internationally. International Meteor Organization (IMO) [11] Video Meteor Network, established by Germany, currently consists of 88 cameras operated by 49 observers from 16 countries. They use MetRec to capture meteor tracks from video sequences at the standard frame rates of 25 or 30 FPS of the cameras. SonotaCo Network, located in Japan, consists of 70 observation stations. It utilizes stars for field calibration and employs software such as UFOCapture, UFOAnalyzer and UFOOrbit to detect and analyze meteors. The Cameras for All-sky Meteor Surveillance (CAMS) network [12] consists of three observation stations, with each station operating 60 cameras. It can monitor the sky at altitudes above 31 degrees. MeteorScan-based trajectory search technology autonomously processes meteor images acquired from each station. European viDeo MeteOr Network Database (EDMOND) [13] is generated through collaboration and data sharing among several European national networks and IMO Video Meteor networks. It collects data from 155 sites and utilizes UFOOrbit for automated processing of meteors. The nine-channel Mini-Mega TORTORA (MMT-9) [14] was deployed in 2014 at a specialized astrophysical observatory near the 6 m Russian telescope. It offers high time-resolution with exposure times ranging from 0.1 to several hundred seconds. It has the capability to observe a wide range of 900 square degrees or a narrow range of 100 square degrees of the sky simultaneously, enabling real-time data processing and effective detection of various transient events. This system has detected over 90,000 meteors. Meteors occur at a rate of 300–350 per night, with durations ranging from 0.1 to 2.5 s and angular velocities reaching up to 38 degrees per second. The Canadian Automated Meteor Observatory (CAMO) [15] is an automated video meteor system comprising two sites. It comprises two enhanced cameras: a wide-field camera with a 28° field of view, utilized for collecting meteor light curves and calculating trajectories; and a narrow-field camera with a 1.5° field of view, employed for tracking meteors and obtaining high-resolution observations. The cameras have a resolution of $640 \times 480$, with the wide-field camera capturing 80 frames per second and the narrow-field camera capturing 110 frames per second. ASGARD is used for real-time detection of the wide-field images to guide the narrow-field tracking of meteors.

GWAC [16–18], established at the Xinglong Observatory of the National Astronomical Observatories of China, is used for the measurement of optical transient objects before, during, and after gamma-ray bursts [19]. It consists of 40 joint field of view (JFoV) cameras. Each camera, with an aperture of 18 cm and custom-made lenses with an f-ratio of f/1.2, is equipped with a $4k \times 4k$ E2V back-illuminated CCD chip. The wavelength range is from 0.5 to 0.85 μm. The field of view for each camera is 150 $\text{deg}^2$ and the pixel scale is $11''7$. The total field of view for each unit carrying four JFoV cameras is ~600 $\text{deg}^2$. The entire camera array covers an area of 5000 $\text{deg}^2$ in the sky, capturing an image every 15 s, including a 10-s exposure and 5-s readout [20]. The limiting magnitude reaches a V-magnitude of 16, allowing for the detection of various moving objects such as meteors, asteroids, comets, airplanes, space debris, and satellites. The maximum brightness value before the system reaches saturation is 8 mag. Continuous observations are conducted each observing night for up to 10 h on specific areas of the sky. The GWAC system is designed for short-timescale sky surveys, aiming to detect accurately and rapidly astronomical anomalies. The detection of meteors is a significant component of the GWAC system, offering valuable insights into meteor trajectories, brightness levels, and mass distribution information.

International meteor surveillance systems are characterized by multi-site observations, primarily using the video, and focusing on brighter meteors [21]. However, the GWAC system differs in the following aspects: (1) It lacks the ability to measure the height and

velocity information of moving objects through single-site observations. (2) It captures images at a frequency of every 15 s, with a resolution of 4k × 4k. While meteor duration typically ranges from 0.1 to 2.5 s, so it often appears as single frames in GWAC system. (3) There is a presence of a significant amount of background moving objects. The current international meteor detection algorithms are not applicable for the meteor detection in the GWAC system. Xu Yang and colleagues [22] conducted a study on identifying meteor candidates using two months of data from the mini-GWAC system in 2019. The mini-GWAC system has a lower image resolution of only 3k × 3k and captures fewer background stars and moving objects. They provide data indicating that aircraft can achieve angular velocities of 1.85 degrees per second and satellites can attain angular velocities of 0.025 degrees per second with durations of 10 s. This enables them to remain within the image frame for an extended period before exiting. As a result, they exhibit continuous presence across multiple frames. We exclude non-meteor objects by considering their presence in multiple frames. Consequently, using this algorithm for meteor candidate detection for the GWAC system resulted in several issues, including missing and false detection of moving objects, incomplete clustering of moving objects, and low precision in single-frame moving-object filtering. Therefore, we presented a meteor detection algorithm specifically designed for the GWAC system based on the algorithm proposed in Reference [22]. Our algorithm can detect meteors from GWAC images with a magnitude range of about −0.66~−7.26 mag. For the brightest meteor, the GWAC can only record a part of it, so it should be brighter than the magnitude given by the GWAC. Our proposed algorithm aims to achieve real-time meteor detection while improving accuracy.

The main contributions of this study are as follows: (1) to apply image block division to set image binarization threshold and introduce median filtering to reduce noise and detect more line segments, avoiding the issues of false detection and missed detection of numerous moving objects. (2) to utilize the origin moment information of the line segment in the moving objects clustering model, and we employ the line segment inclination angle information to verify the credibility of the clustering objects to improve the accuracy of moving objects clustering. (3) to propose a moving object tracking algorithm. The concept of the maximum disappearance frame for moving objects introduced in the moving objects tracking algorithm contributes to tracking multi-frame moving objects. The inclination angle of the line segments of moving objects utilized as the direction of movement facilitates the tracking of multiple moving objects. Thereby it can improve the probability of correctly filtering single-frame moving objects and the accuracy of meteor detection. (4) to conduct comparative experiments between our proposed algorithm and the Reference [22] algorithm to validate the effectiveness. Our algorithm achieves a precision of 89.4% and an improvement of 8.4%, resulting in an average processing time of 0.39 s per image frame. Additionally, we utilized data collected by GWAC system from 10th to 19th December in 2019 and 2021 for meteor detection. The precision of our algorithm was found to reach 90.27%.

## 2. Materials and Methods

This study utilizes data collected from the GWAC system and proposes a new meteor detection algorithm based on the mini-GWAC system algorithm. In the first step, we apply the temporal difference method to remove non-transient objects from adjacent frame images. Subsequently, we employ the PHT to identify the line segments corresponding to the detected moving objects after conducting image preprocessing. A moving objects clustering algorithm is applied in order to cluster these line segments. In the second step, we implement a single-frame moving object tracking algorithm that utilizes the angle feature of the line segments to effectively filter single-frame moving objects. Finally, by analyzing the light curves of the single-frame moving objects, we can classify the single- and double-peaked single-frame moving objects as potential meteor candidates. The flowchart illustrating our meteor detection algorithm is presented in Figure 1.

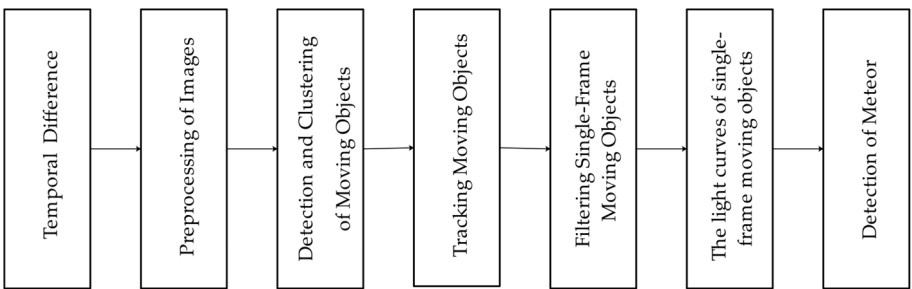

**Figure 1.** Flowchart of our meteor detection algorithm.

*2.1. Dataset*

We collected data from the GWAC system during the Gemini meteor shower period, including seven days from 12 December to 18 December 2021, and eight days from 10 December to 14 December and 17 December to 19 December 2019. The collected data for each date are shown in Figure 2.

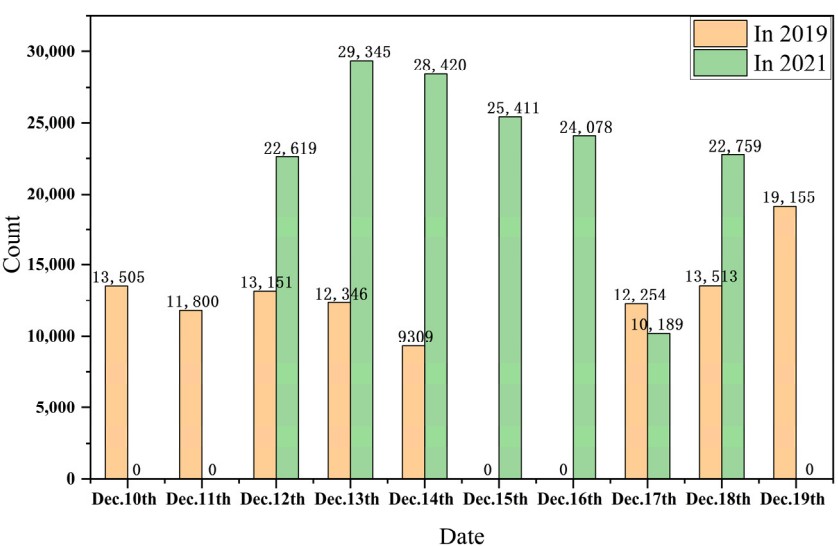

**Figure 2.** Raw events of experimental data for each date in 2019 and 2021.

The GWAC system has a time sampling period of 15 s, and each camera can capture a maximum of 240 images per hour. It captures the fixed sky area for approximately 10 h during each observing night. From 10 December to 19 December 2019, a total of 105,033 images were collected by the GWAC system. Similarly, the GWAC system captured a total of 162,821 images from 12 December to 18 December 2021. Consequently, the total number of images to test obtained in 2019 and 2021 was 267,854.

*2.2. Meteor Detection Model*

2.2.1. Temporal Difference

Temporal difference [23] is a widely employed technique for detecting moving objects, applied diverse applicability in various domains including video analysis, video surveillance, and action recognition. It detects moving objects by subtracting the pixel values between adjacent frames.

$$D_t(x, y) = I_t(x, y) - I_{t-1}(x, y) \tag{1}$$

where $I_{t-1}(x, y)$ represents the pixel values of the $t-1$ frame image, $I_t(x, y)$ represents the pixel values of the $t$ frame image, and $D_t(x, y)$ represents the pixel differences between the adjacent and $t-1$ frame images. In order to identify transient moving objects, it is

used to remove stars, filter out sporadic noise and cosmic rays to reduce the influence of background moving objects.

### 2.2.2. Preprocessing of Images

Preprocessing operations need to be applied to the differenced images before recognizing moving objects. Image preprocessing includes image binarization and median filtering. Firstly, the differenced images are subjected to binary thresholding. The images captured by the GWAC system have a high resolution of 4k × 4k. The binary thresholding is set as twice the standard deviation of the average pixel across the entire image. There is a risk of losing potential moving objects if the threshold is set too high in cases of bright backgrounds. Likewise, setting the threshold too low in cases of dark backgrounds can result in substantial background noise. Therefore, it is divided into 2 × 2 small images, resulting in four regions $d_i(i = 1, 2, 3, 4)$. The threshold $T_i$ for each region is calculated as $T_i = \mu_i + 2\sigma_i(i = 1, 2, 3, 4)$, where $\mu_i$ and $\sigma_i{}^2$ represent the average pixel intensity and variance of each region, respectively. The formula for image binarization is as follows:

$$d_i(x,y) = \begin{cases} 255 & d_i(x,y) > T_i \\ 0 & \text{others} \end{cases} (i = 1, 2, 3, 4) \qquad (2)$$

Pixels above the threshold are assigned the color white, while pixels at or below the threshold are assigned the color black. Utilizing an adaptive block-wise adjustment of the binary thresholding allows for a more precise differentiation between moving objects and their background, resulting in reduced background noise and accurate preservation of the detailed trajectories of the moving objects. Subsequently, the binary images undergo median filtering to reduce noise and prevent interference with PHT for detection. Recognizing that moving objects usually cover multiple pixels, a 3 × 3 template is deliberately chosen for median filtering to effectively remove isolated noise points.

### 2.2.3. Detection and Clustering of Moving Objects

The differenced images are utilized to distinguish the moving objects from the background following image preprocessing. The moving objects appear as straight lines, which can be recognized using HT. Hough transform [24] is based on the concept of mapping a pixel point $(x, y)$ from the image space to a sinusoidal curve $r = x \cos \theta + y \sin \theta$ of the polar coordinate space $(r, \theta)$. The sinusoidal curve represents all possible straight lines passing through that point. Collinear points in the image space are represented by a single point $(r', \theta')$ at which sinusoidal curves in the polar coordinate space intersect. Straight lines can be detected by evaluating the accumulation value at that point $(r', \theta')$ in the polar coordinate space, as illustrated in Figure 3. HT can also be applied to detect geometric shapes such as circles, ellipses, and other specific shapes in addition to line detection [25,26].

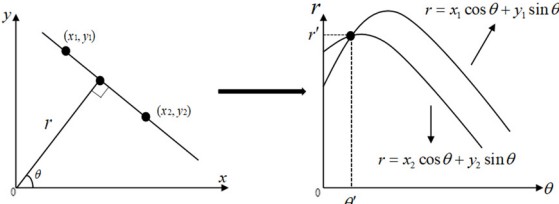

**Figure 3.** Points in the image space to sinusoidal curves in the polar coordinate space.

HT exhibits high computational complexity, particularly when handling large-sized images. The PHT is employed as a solution to mitigate this problem by reducing the computational burden. A subset of randomly selected points from the image is considered as sample points in the PHT, which then finds other connected sample points and accumulates votes based on their relationships to determine the parameters of the lines. The linear

features and positions of the moving object can be extracted by utilizing the computation and analysis of the PHT [27].

The PHT recognizes moving objects, generating multiple line segments forming a set. Hence, the clustering of line segments from the moving objects is an essential step. The process of clustering line segments utilizes two primary features [28]: the line segment origin moment $d$ and the inclination angle $\theta$. The implementation method of line segments clustering is as follows:

1. The new clustering object set is initialized by adding the origin moment $d$, the average origin moment $\overline{d}$, two position coordinates $(x_1, y_1)$ and $(x_2, y_2)$, the inclination angle $\theta$, the average of the inclination angle $\overline{\theta}$ and the variance of the inclination angle $\hat{\theta}$ from the first line segment or unmatched line segment in the line segments set.

$$\Delta y = y_2 - y_1, \Delta x = x_2 - x_1 \tag{3}$$

$$\theta = \arctan\frac{\Delta y}{\Delta x} \tag{4}$$

$$l = \sqrt{(\Delta x)^2 + (\Delta y)^2} \tag{5}$$

$$d = \frac{|y_1 \Delta x - x_1 \Delta y|}{l} \tag{6}$$

2. The distance is calculated between the origin moment $d$ of the line segment in the line segment set and the average origin moment $\overline{d}$ of each clustering object set in turn. If the distance is less than the maximum distance error (MDE), the line segment is classified into the same clustering object set, $\overline{d}$, $\overline{\theta}$ and $\hat{\theta}$ are updated in clustering object set. If the line segment is matched with all clustering object sets, step 1 is executed to add a new clustering object set.

3. When all the line segments in the line segment set are clustered, the clustering reliability of each clustering object set is judged by $\hat{\theta}$ of the line segments within each clustering object set.

4. If the variance of the inclination angles $\hat{\theta}$ of the clustering object set is less than the maximum inclination angle error (MIAE), it indicates a successful clustering set; otherwise, it fails. Then the inclination angle error is calculated between the inclination angle $\theta$ of the line segments of the failed clustering object set and the average inclination angle $\overline{\theta}$ of each clustering object set in turn. If it is less than MIAE, this line segment is matched with this clustering object set; otherwise, step 1 is executed to add a new cluster object set.

5. Each clustering object set represents a moving object. The longest line segment in the clustering object set is denoted as the longest trajectory of the moving object, with position coordinates $(x_{min}, y_{min})$ and $(x_{max}, y_{max})$.

### 2.2.4. Tracking Moving Objects and Filtering Single-Frame Moving Objects

Meteor characteristics in the GWAC system primarily appear in single frames, while non-meteor objects tend to appear in multiple frames. We track moving objects in each frame to detect if the moving objects in the current frame match with the previous ones, consequently filtering single-frame moving objects [29]. The main implementation method is as follows:

6. The position coordinates $(x_{min}, y_{min})$ and $(x_{max}, y_{max})$, the inclination angle $\theta$, and midpoint coordinate $(x_{mid}, y_{mid})$ of each moving object in every frame are obtained. If it is the first frame, each moving object is assigned an initial ID.

7.　It is firstly necessary to determine whether the moving objects in the current frame match with the ones in the previous frames.

8.　We calculate the inclination angle $\theta'$ of the line segments formed by pairing the midpoints $(x_{mid}, y_{mid})$ of the moving objects in the current frame with the midpoints $(x_{obj}, y_{obj})$ of the moving objects in the tracking object set, and create a matrix $M$ of the inclination angles $\theta'$.

$$M(i,j) = \arctan\frac{y_{mid}(i) - y_{obj}(j)}{x_{mid}(i) - x_{obj}(j)}(i = 1,2,3,\ldots,m; j = 1,2,3,\ldots,n) \qquad (7)$$

9.　Where the total of moving objects in the current frame is $m$, while the tracking object set consists of $n$ moving objects.

10.　We update the ID and the position coordinate of the moving objects in the current frame. The inclination angle $\theta_j$ of the moving object $j$ in the tracking object set indicates its movement direction in the next frame. The inclination angles $\theta'_{i,j}$ in column $j$ of the matrix $M$ indicate the actual movement directions of the moving object $j$. Then we find the moving object $i$ in the current frame that inclination angle $\theta'_{i,j}$ is the nearest $\theta_j$ and calculate the error in the inclination angle between the two. If this error is less than MIAE, the moving object $i$ in the current frame is successfully matched the moving object $j$ in the tracking object set. They are regarded as the same object and assigned the same ID. The appearance count of this moving object $j$ is added by 1.

11.　If there are moving objects matched unsuccessfully in the tracking object set, they are marked as disappeared moving objects. When the number of frames in which they have disappeared exceeds the maximum disappearance frame (MDF), they are removed from the tracking object set.

12.　If there are moving objects matched unsuccessfully in the current frame, they are marked as newly appeared moving objects and assigned a new ID. They are added to the tracking object set along with their position coordinates and inclination angles. The appearance count of these objects is incremented by 1.

13.　We filter single-frame moving objects. When the moving object is marked as disappeared and the number of frames it has been missing exceeds the MDF, it is considered a single-frame moving object if the appearance count of it is 1.

In the moving-object tracking algorithm, the concept of the MDF for moving objects introduced in the moving objects tracking algorithm is contributed to track multi-frame moving objects. The inclination angle of the line segments of moving objects utilized as the direction of movement facilitates the tracking of multiple moving objects. Thereby it can reduce the probability of incorrectly filtering single-frame moving objects and improve the accuracy of meteor detection.

### 2.2.5. Meteor Detection

In addition to utilizing the meteor characteristics in the GWAC system appearing in a single frame, meteor features can also be used for further filtering single-frame moving objects to select meteor candidates and improve the accuracy of meteor detection. Meteors typically exhibit a brief bright light and a long tail, distinguishing them from other moving objects such as airplanes and satellites, as shown in Figure 4.

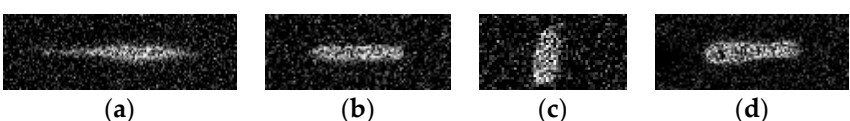

(**a**)　　　　　　(**b**)　　　　　　(**c**)　　　　　　(**d**)

**Figure 4.** Different moving objects observed for the GWAC system: (**a**) meteor; (**b**–**d**) other moving objects.

The variation of brightness of celestial bodies over time is referred to as the light curve in astronomy [30]. It is not possible to obtain the variation curve of brightness of moving

objects over time in the GWAC system. Therefore, the variation curve of brightness of moving objects with respect to the image pixels is used for the light curve and defined as the trajectory of the moving objects [22]. By calculating the light curves of single-frame moving objects, we analyze their shapes, and filter out non-meteor single-frame moving objects to obtain meteor.

- Preprocessing of single-frame moving objects

The images are rotated clockwise $\alpha$ to obtain horizontally moving objects. The rotation transformation is as follows:

$$\left(x' \; y' \; 1\right) = \left(x \; y \; 1\right) \begin{pmatrix} 1 & 0 & 0 \\ 0 & -1 & 1 \\ -0.5w & 0.5h & 1 \end{pmatrix} \begin{pmatrix} \cos\alpha & -\sin\alpha & 0 \\ \sin\alpha & \cos\alpha & 0 \\ 0 & 0 & 1 \end{pmatrix} \begin{pmatrix} 1 & 0 & 0 \\ 0 & -1 & 1 \\ 0.5w' & 0.5h' & 1 \end{pmatrix} \quad (8)$$

where the co-ordinates of the image before rotation are $(x, y)$, and after rotation they are $(x', y')$. $w$ and $h$ represent the width and height of the image before rotation. $w'$ and $h'$ represent the width and height of the image after rotation. The rotation angle $\alpha$ is $180° - \theta$. The position co-ordinates of the single-frame moving object before rotation are $(x_{min}, y_{min})$, $(x_{max}, y_{max})$, and after rotation they are $(x'_{min}, y'_{min})$ and $(x'_{max}, y'_{max})$. The rotation angle of the moving object is $\theta$. Next, the region is cropped, and the cropping area is as follows.

$$area(x, y) = \begin{bmatrix} (x'_{min} - 30, \; y'_{min} - 10) & \cdots & (x'_{max} + 30, \; y'_{min} - 10) \\ \vdots & \ddots & \vdots \\ (x'_{min} - 30, \; y'_{min} + 10) & \cdots & (x'_{min} - 30, \; y'_{min} + 10) \end{bmatrix} \quad (9)$$

- The light curves of single-frame moving objects

We sum the brightness along the vertical axis with respect to the horizontal axis in the cropping area of the single frame moving object, resulting in the light curve.

$$F(x) = \sum_{y} area(x, y) \quad (10)$$

The light curve exhibits abundant spike noise. To reduce the noise, we perform a smoothing convolution, resulting in smoothed light curves. We then detect the peaks in the light curves and classify them into flat peaks, single-peaked, or double-peaked moving objects based on the number of peaks. This is illustrated in Figure 5.

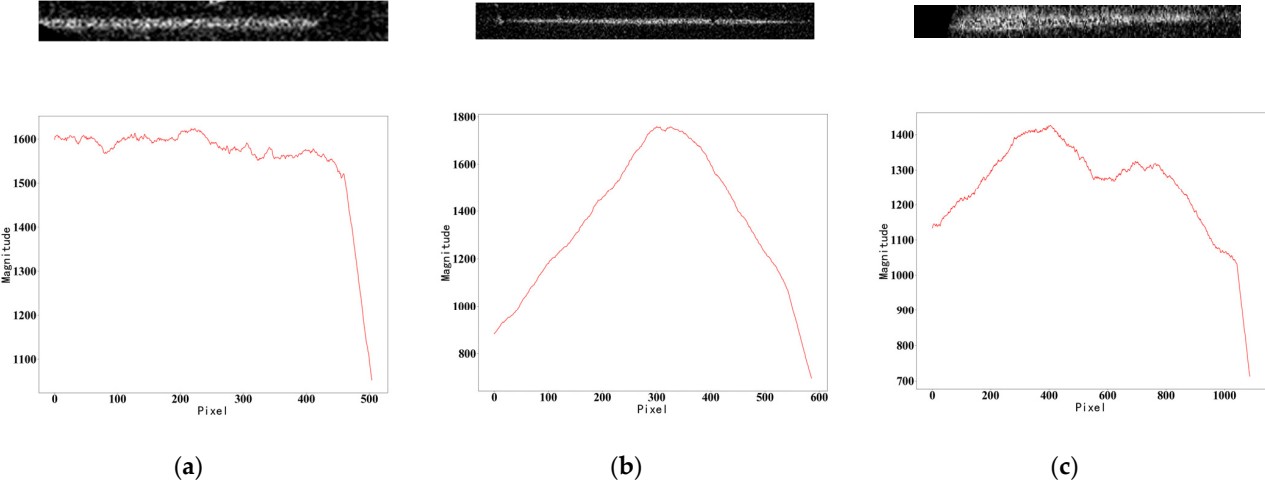

**(a)**    **(b)**    **(c)**

**Figure 5.** The light curves of single-frame moving objects in the GWAC system: (**a**) A flat peak object and its light curve; (**b**) A single-peaked object and its light curve; (**c**) A double-peaked object and its light curve.

According to [30], flat curves of meteors only account for 15% of the total classified objects in meteor events. The category of flat curves comprises a small number of meteors but large number of non-meteor objects. To obtain highly accurate meteor candidates, single-peaked and double-peaked moving objects are selected, filtering out non-meteor moving objects.

## 3. Results and Discussions

### 3.1. Implementation Details

The meteor detection algorithm was implemented using Python and OpenCV. The experiments were conducted on an AMD Ryzen 7 4800H CPU and a NVIDIA GeForce GTX 1650 Ti GPU.

The parameters for PHT are established as follows: $\rho$ is set to one pixel, the accumulator threshold parameter is set to 35, the minimum length of line segments is set to 35 pixels, and the maximum allowed gap between line segments is set to 10 pixels. As for line-segment clustering, the maximum distance error is limited to 200 pixels, and the maximum inclination angle error is constrained at 3°. As for moving objects tracking, it is ensured that the maximum disappearance frame are kept at 3.

The results of the object detection are evaluated using precision (P) and recall (R). Precision represents the probability of correctly identifying meteor samples among the data identified as meteor samples. Recall represents the probability of correctly identifying meteor samples among meteor samples in the dataset.

$$\text{P} = \frac{TP}{TP + FP} \tag{11}$$

$$\text{R} = \frac{TP}{TP + FN} \tag{12}$$

where *TP* represents true positive, *TN* represents true negative, *FP* represents false positive, and *FN* represents false negative.

### 3.2. Experimental Results

#### 3.2.1. Comparative Experiments

To provide a visual and accurate comparison between our proposed algorithm and the algorithm proposed in Reference [22], Figure 6 displays the results of temporal difference and image preprocessing for both algorithms. From the comparison shown in Figure 6a,b, moving objects are not observed. Figure 6c shows the presence of overexposed regions after temporal difference of the Reference [22] algorithm. In Figure 6d, image preprocessing does not handle the regions, resulting in false detections in the PHT. In contrast, it can be observed that our proposed algorithm does not have overexposed regions after the temporal difference and image preprocessing from Figure 6e,f. The noise is significantly reduced, avoiding false detection of moving objects.

The results of image preprocessing and PHT for both the Reference [22] algorithm and our proposed algorithm are showed in Figure 7. The comparison between Figure 7a,b reveals the presence of transient moving objects in adjacent frames. After applying the image preprocessing of the Reference [22] algorithm, a significant amount of noise interferes with the subsequent recognition for the PHT in Figure 7c, leading to the absence of detected moving objects in Figure 7d. However, the image preprocessing with block-based thresholding and median filtering significantly reduces noise in Figure 7f, resulting in the detection of moving objects in Figure 7e using the PHT. Therefore, the comparative experiments demonstrate that block-based thresholding for image binarization and median filtering can effectively reduce noise and prevent the occurrence of missed detection of moving objects.

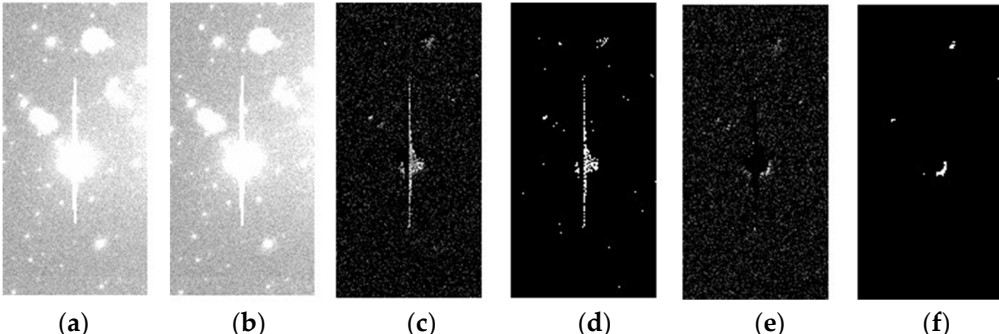

(**a**) (**b**) (**c**) (**d**) (**e**) (**f**)

**Figure 6.** The results of the adjacent frame temporal difference and image preprocessing: (**a**) A partial view of the G024_Mon_objt_211214T20201965 (frame t); (**b**) A partial view of the G024_Mon_objt_211214T20203464 (frame t + 1); (**c**,**d**) The results of the Reference [22] algorithm; (**e**,**f**) The results of our proposed algorithm.

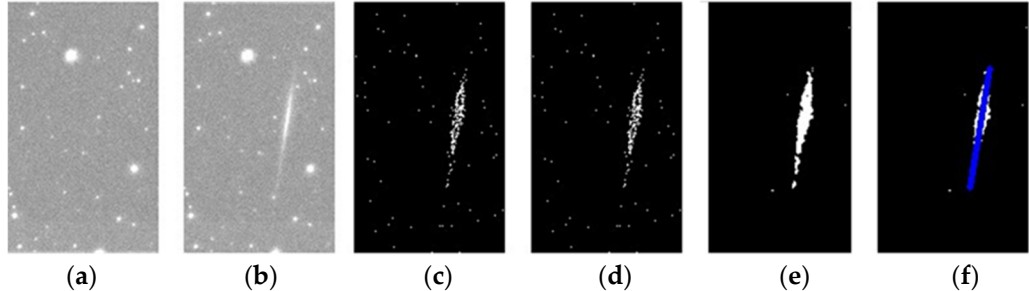

(**a**) (**b**) (**c**) (**d**) (**e**) (**f**)

**Figure 7.** The results of the adjacent frame image preprocessing and PHT: (**a**) A partial view of the G024_Mon_objt_211214T19104960 (frame t); (**b**) A partial view of the G024_Mon_objt_211214T19110460 (frame t + 1); (**c**,**d**) The results of the Reference [22] algorithm; (**e**,**f**) The results of our proposed algorithm.

The results of the line segments clustering for both the Reference [22] algorithm and our proposed algorithm are displayed in Figure 8. From Figure 8a, it can be observed that there is only one moving object in the image G024_Mon_objt_211214T20281960. However, Figure 8b and its bottom-left magnified portion show that the Reference [22] algorithm clusters this moving object into two separate ones. Figure 8c demonstrates the effectiveness of our proposed algorithm in clustering the moving object into a single object.

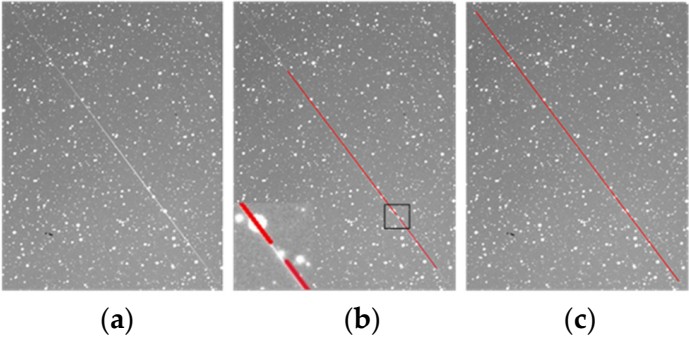

(**a**) (**b**) (**c**)

**Figure 8.** The results of line segments clustering: (**a**) A partial view of the G024_Mon_objt_211214T20281960; (**b**) The clustered line segments by the Reference [22] algorithm; (**c**) The clustered line segments by our proposed algorithm. **Note:** The red lines represent the clustering results, where each line corresponds to a clustering moving object.

The Reference [22] algorithm only considers the tracking of adjacent frames for moving objects. When a moving object appears in adjacent frames and is missed in a certain frame, there is possibility of mistakenly identifying the last frame's moving object as a single-frame moving object. As shown in Figure 9a–c, when a moving object appears in three adjacent frames, it is labeled as ID:0 in the first frame. If this moving object is not detected in the second frame, resulting in the failed match, the first frame of this moving object is erroneously classified as a single-frame moving object. This, in turn, affects the accuracy of meteor detection. Our proposed algorithm sets the MDF as three times, meaning that object can still be matched with the same object even after disappearing for three frames. As shown in Figure 9d–f, when a moving object appears in the first frame with ID:0 and is matched with the moving object in the second frame, if the second frame doesn't detect this moving object resulting in a failed match, the position co-ordinates of the moving object from the first frame are retained, and it is considered as one disappearance of the object. Later, when matched with the moving object in the third frame, it is labeled as ID:0, accurately identifying the single-frame moving object. Since it reappears several frames later, we can exclude that this was a meteor.

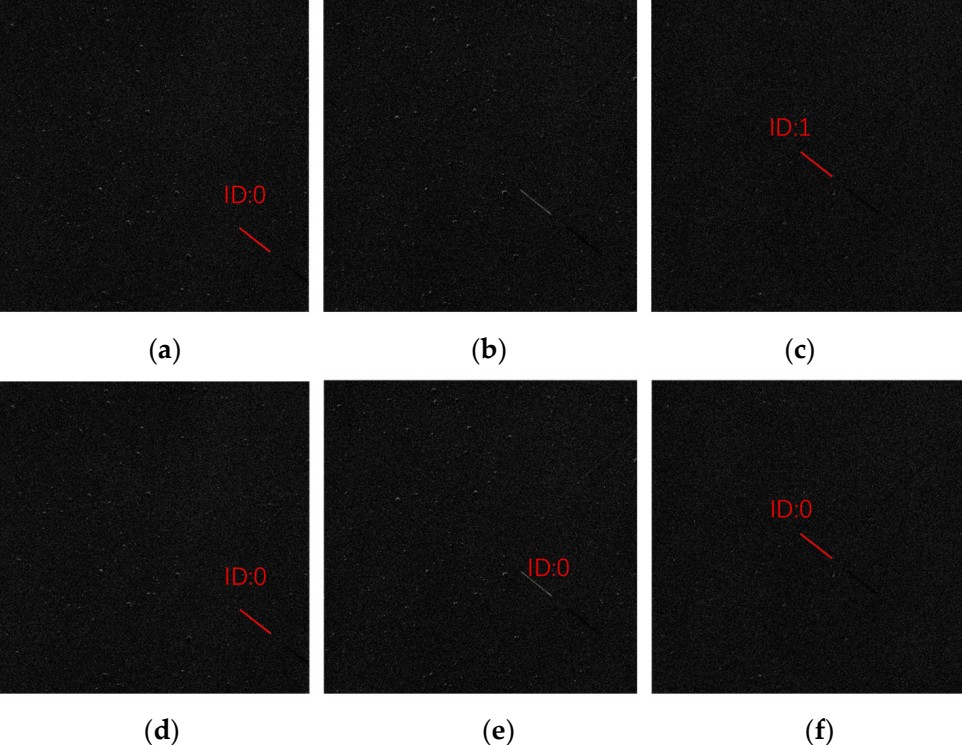

**Figure 9.** The tracking results of multi-frame moving object: (**a**–**c**) The tracking results of multi-frame moving object by the Reference [22] algorithm; (**d**–**f**) The tracking results of multi-frame moving object by our proposed algorithm.

The Reference [22] algorithm only utilizes the inclination angle of adjacent line segments, which leads to the problem of matching the last frame's moving objects with multiple moving objects in the current frame as the same object. As shown in Figure 10a,b, the inclination angle of the moving object in the first frame is 154.18°, while in the second frame it is 153.86°. The inclination angle of the new moving object (left side in Figure 10b or Figure 10d) is 156.98°. The Reference [22] algorithm assigns ID:0 to the moving object in the first frame, and all moving objects in the second frame that have the inclination angle within 5° difference from the first frame's moving object are matched, resulting in mistakenly identifying the newly appeared moving object in Figure 10b as the same moving object. Our proposed algorithm, on the other hand, utilizes the inclination angle of the line segments from the first frame's moving object as the reference for the movement

direction of the next frame's moving object. The inclination angle between the midpoint of the second frame's (current frame) moving object and the midpoint of the previous frame's moving objects is considered as the direction of the actual trajectory. If the difference in the direction inclination angle is less than 3°, they are matched as the same object. As shown in Figure 10d, the moving object in the second frame is successfully matched with the moving object in the first frame, with ID:0, and a new moving object with ID:1 is detected.

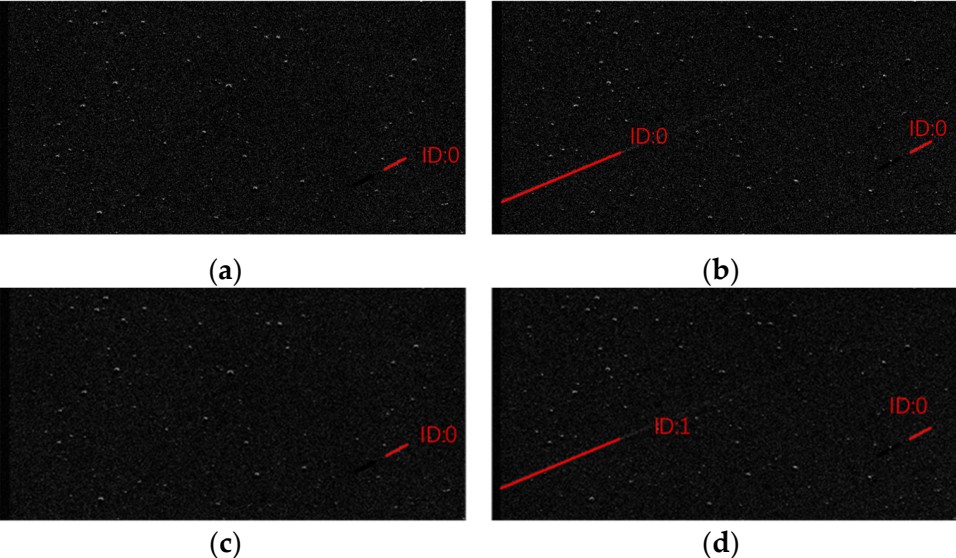

**Figure 10.** The tracking results of multiple moving objects: (**a**,**b**) The tracking results of multiple moving objects by the Reference [22] algorithm; (**c**,**d**) The tracking results of multiple moving objects by our proposed algorithm.

The visual results of the experimental comparison between our proposed algorithm and the Reference [22] algorithm from Figures 6–10 demonstrate our proposed algorithm can effectively reduce noise, prevent missed detections of moving objects, accurately identify the single-frame moving object and track multiple moving objects. This validates the effectiveness of our algorithm.

In this study, 5856 adjacent frames images from 14 December 2021 were selected as the dataset for accurate comparison. Single-frame moving objects were filtered and further refined to select meteor candidates. The selected dataset was manually annotated, revealing a total of 242 single-frame moving objects and 189 meteor candidates. The algorithms proposed by Reference [22] and this study were used to filter the single-frame moving objects, and precision and recall were calculated for each algorithm. In addition, there are a few edge objects in the single-frame moving objects. After removing the edge objects, the Reference [22] algorithm and our proposed algorithm were used to detect meteor candidates. Similarly, precision and recall were calculated. The calculation results are shown in Table 1. The precision of our proposed algorithm in single-frame moving object selection reaches 91.4%, which is an improvement of 13.3% compared to the Reference [22] algorithm. Out of 242 single-frame moving objects, 88.0% were correctly filtered. In terms of meteor selection, our algorithm achieves a precision of 89.4%, showing an improvement of 8.4% compared to the Reference [22] algorithm. Out of 189 meteor objects, 83.5% were accurately detected. The high precision of the meteor detection algorithm ensures the high probability of correctly identifying meteor samples among the data identified as meteor samples, which provides valuable positional information for further research on the brightness and mass of meteors.

**Table 1.** The precision and recall for filtering single-frame moving objects and detection of meteors by the Reference [22] algorithm and our proposed algorithm.

| Method | Filtering Single-Frame Moving Objects | | Detection of Meteors | |
|---|---|---|---|---|
| | Precision | Recall | Precision | Recall |
| The Reference [22] | 0.781 | 0.759 | 0.814 | 0.793 |
| Our algorithm | 0.914 | 0.880 | 0.898 | 0.835 |

The time characteristic of the entire meteor detection process was evaluated by comparing the Reference [22] algorithm with our proposed algorithm on the dataset of 5856 image frames. The summarized experimental results can be found in Table 2. The Reference [22] algorithm required 6965 s to process the 5856 image frames, resulting in an average processing time of 1.19 s per image frame. In contrast, our proposed algorithm completed the process in 2280 s, resulting in an average processing time of 0.39 s per image frame. This indicates an average improvement of 0.8 s per image frame compared to the Reference [22] algorithm, satisfying the real-time requirements of the GWAC for meteor detection.

**Table 2.** The time of the entire meteor detection process by the Reference [22] algorithm and our proposed algorithm.

| Method | The Reference [22] | Our Algorithm |
|---|---|---|
| Time/s | 6965 | 2280 |

### 3.2.2. Meteor Detection for the GWAC Data

Firstly, moving-object identification was conducted on 267,854 images from 2019 and 2021. The results showed that 11,460 moving objects in 2019 and 26,523 moving objects in 2021, totaling 37,983 moving objects, were identified. Subsequently, the identification of meteor objects required further selection of single-frame moving objects. The filtering results revealed that 2759 single-frame moving objects were selected in 2019 and 6742 single-frame moving objects were selected in 2021, with a total of 9501 single-frame moving objects. In 2019, single-frame moving objects accounted for 24.08% of the total number of moving objects, while in 2021 they accounted for 25.41%. The overall selection of single-frame moving objects in 2019 and 2021 accounted for only 24.75% of the total number of identified moving objects, indicating that non-meteor objects such as airplanes and satellites with multiple frames constituted a higher proportion of the total moving objects. The results of total identified moving objects and single-frame moving objects for each date in 2019 and 2021 are presented in Figure 11. On 13 December 2021, during the peak period of the Geminid meteor shower, the highest number of single-frame moving objects was identified, reaching up to 1350. Near the peak period on 14 December 2021, 1224 single-frame moving objects were identified.

Finally, meteor detection was performed on the single-frame moving objects from 2019 and 2021, obtaining highly accurate meteor candidates. The detection of meteors for each date in 2019 and 2021 is presented in the following Table 3.

There were 11.28% edge-moving objects detected among the selected single-frame moving objects in 2019 and 2021. These edge-moving objects only represent partial trajectories, making them indeterminate, and therefore classified as flat-peak moving objects. In 2019, a total of 1719 meteor candidates were selected, out of which 1544 were confirmed through manual verification, resulting in an accuracy of 89.82%. Similarly, 4295 meteor candidates were selected in 2021, with 3897 confirmed through manual verification by checking their light curves, resulting in an accuracy of 90.73%. The average precision in meteors detection for 2019 and 2021 reached 90.27%, with meteor candidates accounting for 15.83% of the total moving objects. From Table 3, it is evident that on December 13th in

2019 and 2021, the Gemini meteor shower reached its peak, with the highest number of detected meteors.

**Table 3.** Number of single-frame moving objects and meteors for each date in 2019 and 2021.

| Date | Number of Single-Frame Moving Objects (2019/2021) | Number of Meteors (2019/2021) | Manually Verified Number of Meteors (2019/2021) |
|---|---|---|---|
| 10 December | 358/0 | 208/0 | 179/0 |
| 11 December | 331/0 | 228/0 | 197/0 |
| 12 December | 370/938 | 216/621 | 201/562 |
| 13 December | 393/1350 | 291/857 | 267/785 |
| 14 December | 212/1224 | 126/833 | 111/747 |
| 15 December | 0/1020 | 0/711 | 0/657 |
| 16 December | 0/1121 | 0/633 | 0/569 |
| 17 December | 368/406 | 206/323 | 194/298 |
| 18 December | 324/683 | 195/317 | 172/279 |
| 19 December | 403/0 | 249/0 | 223/0 |
| Total | 2759/6742 | 1719/4295 | 1544/3897 |

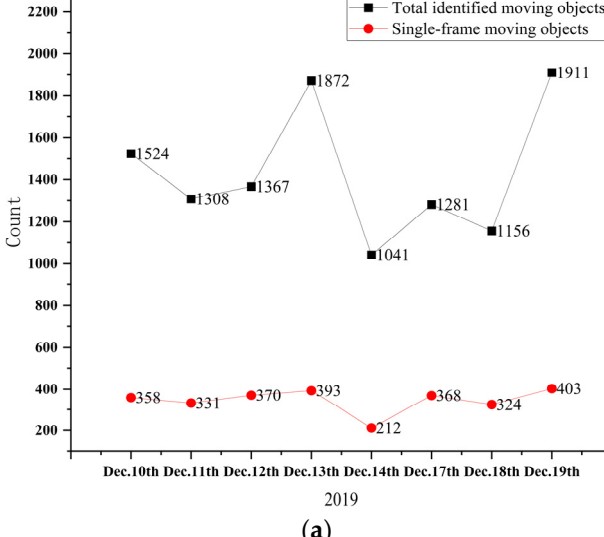

(**a**)

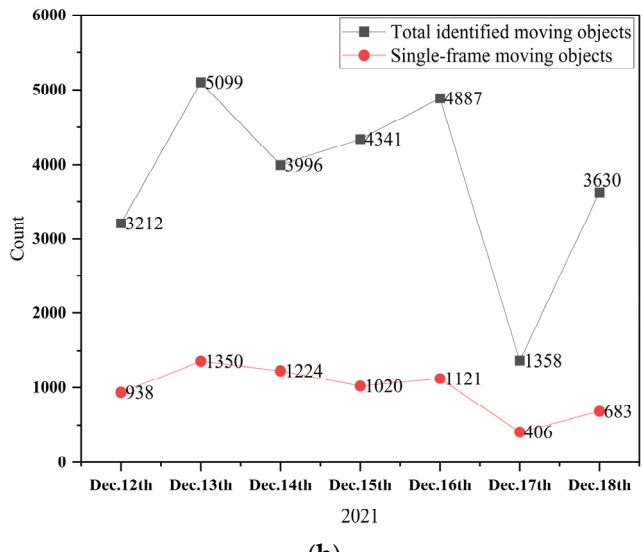

(**b**)

**Figure 11.** The results of total identified moving objects and single-frame moving objects for each date: (**a**) in 2019; (**b**) in 2021.

## 4. Conclusions

The GWAC system captures a significant number of meteors, and the detection of these meteors is crucial for further research on their brightness and mass information. In comparison to international meteor systems, the GWAC system exhibit characteristics such as a V-magnitude of 16, the 150 deg$^2$ field of view for each camera, and single-station observation. These characteristics present challenges for meteor detection in the GWAC system. Current meteor detection algorithms are limited to the mini-GWAC system data, resulting in issues including false detection, missed detection, incorrect object clustering, and mismatched object tracking within the GWAC system. We proposed a meteor detection algorithm applicable to the GWAC system based on the algorithm developed for the mini-GWAC system. Experimental results demonstrate that the proposed algorithm achieves higher accuracy and faster detection speed, with the accuracy of meteor detection reaching up to 90.27%. For future research, we could entail addressing image noise resulting from factors such as camera shake in the GWAC system to further improve the accuracy.

**Author Contributions:** Conceptualization, Y.C. and G.L.; methodology, Y.C. and G.L.; software, Y.C.; validation, Y.C.; formal analysis, Y.C. and G.L.; investigation, Y.C.; resources, G.L.; data curation, G.L.; writing—original draft preparation, Y.C., Q.S. and M.L.; writing—review and editing, G.L., C.L. and B.Q.; visualization, B.Q.; supervision, C.L. and B.Q.; project administration, B.Q.; funding acquisition, B.Q. All authors have read and agreed to the published version of the manuscript.

**Funding:** This research was funded by the Natural Science Foundation of Tianjin, grant number 22JCYBJC00410, and the Joint Research Fund in Astronomy, National Natural Science Foundation of China, grant number U1931134.

**Data Availability Statement:** The data presented in this study are available on request from the corresponding author. The data are not publicly available due to privacy restrictions.

**Conflicts of Interest:** The authors declare no conflict of interest.

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
