# Peer review of "A Meteor Detection Algorithm for GWAC System"

_universe, doi:10.3390/universe9110468_

Round 1

Reviewer 1 Report

Comments and Suggestions for Authors

Comments to Chen et al., "A Meteor Detection Algorithm for GWAC System"20 Sep 2023

In this paper, Chen et al. describe an algorithm to detect meteors using the GWAC (Ground Wide-Angle Camera) system. GWAC was devised for the detection of optical transients after gamma ray bursts. It takes exposures of 10 seconds, followed by 5 seconds readout time. Normally, meteor camera systems record with video frame rate or similar. This allows the fast-moving objects to be easily detected and followed. The authors describe a way to identify meteors in the GWAC data. This would increase the scientific output of the GWAC system, and contribute a lot to meteor science.

The paper describes the algorithm in detail. I do have one fundamental problem in understanding it, which might be easily corrected by giving a clearer overview in the beginning and adding little clarifications here and there. The problem I have is this: Due to the long exposure times, meteors will just show up as a streak of light in one frame. No movement can be detected. Still, the text often talks about moving objects, and puts a lot of emphasis on explaining how objects are tracked from one frame to the next. I would assume that the authors use this to *exclude* that the event is a meteor. But this is not really made clear enough for me.

I have added a lot of little comments in the attached PDF. I hope the authors find it useful. There is a lot of potential in the data, and having a robust way of identifying meteors is important. It could be there - but it should be explained a bit better. It might be that a lot of my comments can be addressed by a clearer introduction in the beginning.

--------------

Comments on the Quality of English Language

Reviewer 2 Report

Comments and Suggestions for Authors

Hello,

This paper present a new and better way to detect meteors out of the images taken by the GWAC cameras. I think the paper is interesting, but lacks the scientific motivation. In short: why are you detecting meteors from GWAC cameras? What is the scientific purpose to develop this method? What do you do with the images?

Other minor comments are included in the pdf.

Reviewer 3 Report

Comments and Suggestions for Authors

The majority of the techniques presented in this paper appear sound and well-founded. Nevertheless, I would appreciate more specific information regarding certain aspects of the parameter settings, such as the chosen threshold for binarization and Hough transformation, among others. Furthermore, I am particularly intrigued by the methodology employed to calculate the recall rate, especially in cases where there may not be an available 'true' catalog for reference.

Comments on the Quality of English Language

The language used in the paper is acceptable. However, it might be beneficial for the authors to consider further improving the quality of their writing by utilizing professional editing services.

Round 2

Reviewer 1 Report

Comments and Suggestions for Authors

Thanks for the updated version and your response letter. I still see a few points I had commented which were not addressed and I kindly ask for one more round of updates. See the attached file for comments.

Round 3

Reviewer 1 Report

Comments and Suggestions for Authors

3rd round of comments to Chen et al., "A Meteor Detection Algorithm for GWAC System"

24 Oct 2023

I am afraid that I still think that some parts of the paper are confusing. Here my comments:

Response 1: thanks for adding the estimated magnitude range of -0.6 to -7 mag. However, then you should not say that 'Meteors captured by the GWAC system are dimmer...' - dimmer than what? Not dimmer than other camera systems. Again, did you confirm that you are not more sensitive?

Response 2: Thanks for adding this. I would use the singular in 'frames' though, i.e. write 'frame'.

Response 3: Unfortunately I still fail to understand some basics. In your cover letter, in response 3, you now write "..Adjacent frames refer to frames within an image sequence that are close to each other in time but may not necessarily have an extremely short time interval between them. These frames may have varying time intervals, which can be seconds, minutes, or even longer. Adjacent frames are often used for capturing events or monitoring situations where a high temporal resolution is not required. Consecutive frames are frames in an image sequence that are captured in quick succession with very short time intervals between them. These frames are captured in a continuous and rapid manner, typically with millisecond or sub-millisecond time intervals." - But: The paper says that images are acquired with exposure times of 10 seconds, readout time 5 seconds. In line 102 you write that it captures images every 15 seconds. So how can you now have data within milliseconds? Please clarify.

Response 4/5: Thanks - I would have preferred a reference to a paper actually giving more details of the camera system. The paper you refer to is where you copied the original text from, the paper we are discussing here is actually already a bit more detailed (explaining what the 'J' in JFoV means, writing that the lenses are custom-made. I would suggest, for future papers based on the system, to publish a technical paper somewhere (e.g. at a conference) giving more of the technical details. For this paper it has to suffice the way it is now.

Response 6: Thanks for the explanation. But then please add in line 499 "... were confirmed through manual verification by checking their light curves, resulting...".

Response 7 - 9: Ok good, thanks.

Author Response

Response 1: thanks for adding the estimated magnitude range of -0.6 to -7 mag. However, then you should not say that 'Meteors captured by the GWAC system are dimmer...' - dimmer than what? Not dimmer than other camera systems. Again, did you confirm that you are not more sensitive?

Response 1:

Thank you for your comments, we have deleted that 'Meteors captured by the GWAC system are dimmer...' in the whole paper. Yes, we confirm it is not more sensitive to detect meteors with a magnitude range of about -0.66 ~ -7.26 mag.

Response 2: Thanks for adding this. I would use the singular in 'frames' though, i.e. write 'frame'.

Response 2:

Thank you for your comments, we have revised the whole paper in 'frame'.

Response 3: Unfortunately I still fail to understand some basics. In your cover letter, in response 3, you now write "..Adjacent frames refer to frames within an image sequence that are close to each other in time but may not necessarily have an extremely short time interval between them. These frames may have varying time intervals, which can be seconds, minutes, or even longer. Adjacent frames are often used for capturing events or monitoring situations where a high temporal resolution is not required. Consecutive frames are frames in an image sequence that are captured in quick succession with very short time intervals between them. These frames are captured in a continuous and rapid manner, typically with millisecond or sub-millisecond time intervals." - But: The paper says that images are acquired with exposure times of 10 seconds, readout time 5 seconds. In line 102 you write that it captures images every 15 seconds. So how can you now have data within milliseconds? Please clarify.

Response 3:

Thank you for your comments, we want to refer to two adjacent images captured by the camera in chronological order . Adjacent frames refer to frames within an image sequence that are close to each other in time. It can be seconds, minutes, or even longer. We have used a method of marking a moving object by differential operation between two adjacent images. So we have used "adjacent frames".

Response 4: Thanks - I would have preferred a reference to a paper actually giving more details of the camera system. The paper you refer to is where you copied the original text from, the paper we are discussing here is actually already a bit more detailed (explaining what the 'J' in JFoV means, writing that the lenses are custom-made. I would suggest, for future papers based on the system, to publish a technical paper somewhere (e.g. at a conference) giving more of the technical details. For this paper it has to suffice the way it is now.

Response 4:

Thank you for your comments.

Response 5: Thanks for the explanation. But then please add in line 499 "... were confirmed through manual verification by checking their light curves, resulting...".

Response 5:

Thank you for your comments, we have revised in line 499.